# Genetic Evolution of Melanoma: Comparative Analysis of Candidate Gene Mutations in Healthy Skin, Nevi, and Tumors from the Same Patients

**DOI:** 10.3390/ijms27010532

**Published:** 2026-01-05

**Authors:** Marta Gil-Barrachina, Barbara Hernando, Gemma Perez-Pastor, Victor Alegre-de-Miquel, Cristian Valenzuela-Oñate, Sandra Minguez-Lujan, Pablo Monfort-Lanzas, Elena Tomas-Bort, Maria Angeles Marques-Torrejon, Conrado Martinez-Cadenas

**Affiliations:** 1Department of Medicine, Jaume I University of Castellon, 12071 Castellon, Spain; marta.gil.barrachina@gmail.com; 2Spanish National Cancer Research Centre (CNIO), 28029 Madrid, Spain; bhernando@cnio.es; 3Department of Dermatology, Consorcio Hospital General Universitario de Valencia, 46014 Valencia, Spain; gemma@aedv.es (G.P.-P.); victor.alegre@uv.es (V.A.-d.-M.); cvalenzuela760@gmail.com (C.V.-O.); sandraminlu@hotmail.com (S.M.-L.); 4Institute of Medical Biochemistry, Medical University of Innsbruck, 6020 Innsbruck, Austria; pablo.monfort@i-med.ac.at; 5Institute of Bioinformatics, Biocenter, Medical University of Innsbruck, 6020 Innsbruck, Austria; 6Centre of Technology and Bioinformatics Millars, 12550 Almazora, Spain; elena@ctbmillars.com

**Keywords:** melanoma, nevus, somatic mutation, clonal progression, melanoma evolution, driver genes

## Abstract

Melanocytic tumorigenesis is thought to occur through stepwise genomic evolution from normal skin to nevi and, ultimately, melanoma. To investigate this progression, we performed targeted deep sequencing of a 46-gene panel in matched healthy skin, nevus, and melanoma samples from 15 patients, including 14 complete tissue trios. Mutation burden increased progressively across tissues, with median mutation counts rising from benign skin to nevi and showing the highest levels in melanoma, consistent with cumulative somatic alterations. Canonical MAPK pathway mutations were common: *BRAF* V600E and *NRAS* Q61 variants were detected in many nevi and melanomas and were shared between lesions in 8 of 15 patients, providing direct evidence of clonal continuity. Variant allele frequencies for driver and nonsynonymous mutations were higher than those of passenger and synonymous mutations, reflecting selective expansion of functionally relevant clones. UV-signature substitutions were abundant, particularly among synonymous variants, suggesting background mutagenesis without clonal advantage. Melanoma-private mutations in genes such as *ARID1A*, *ARID2*, *PIK3CA*, and *CDKN2A* indicated additional late events contributing to malignant progression. Overall, this study supports a model in which many melanomas evolve from pre-existing nevi through sequential acquisition and clonal amplification of somatic mutations, while also revealing heterogeneous evolutionary trajectories.

## 1. Introduction

Cutaneous melanoma is one of the most aggressive forms of skin cancer, with a rising incidence worldwide and significant mortality despite advances in early detection and targeted therapy [1]. The biological origins of melanoma are complex and heterogeneous, involving genetic, environmental, and host factors. Ultraviolet (UV) radiation is the principal environmental mutagen and has been implicated in both the initiation and progression of melanocytic neoplasia [2]. However, the precise evolutionary steps by which normal melanocytes acquire driver mutations, form precursor lesions (e.g., nevi), and eventually progress to invasive melanoma, are still not completely understood.

Over the past two decades, genomic studies have identified recurrent alterations in the MAPK pathway, particularly *BRAF* and *NRAS* mutations, as early and frequent events in melanocytic neoplasia [3,4,5]. Such mutations are common in benign nevi, suggesting that they may initiate clonal expansion but are insufficient for malignant progression on their own [6]. Stark et al. [7] demonstrated that nevi are characterized by limited mutational complexity despite harboring oncogenic *BRAF* or *NRAS* mutations, whereas progression to melanoma requires additional cooperating events affecting tumor suppressors and chromatin regulators. Similarly, Shain et al. [8] used multi-region sequencing of melanomas and their precursor lesions to show a stepwise accumulation of somatic mutations, highlighting the sequential inactivation of *CDKN2A* and disruption of pathways such as p53 and SWI/SNF as critical to tumor evolution.

The mutational landscape of normal sun-exposed skin further complicates this evolutionary trajectory. It turns out that UV-exposed epidermis harbors a surprisingly high load of cancer-associated mutations, including canonical melanoma drivers, even in clinically normal tissue [9,10]. These findings suggest that the background mutational “field” from which nevi and melanomas arise may already be enriched for oncogenic changes, raising questions about the threshold at which such mutations contribute to neoplasia. In contrast, studies such as Curtin et al. [11] emphasized the genetic divergence between melanoma subtypes, supporting a model where different constellations of mutations and environmental exposures shape distinct evolutionary routes.

Controversies remain regarding the timing and sequence of key alterations. While some authors suggest that *BRAF* or *NRAS* mutations are obligatory initiating events [6,7], others have identified melanomas lacking these mutations yet harboring alterations in *NF1* or other MAPK regulators [11]. Similarly, the extent to which precursor lesions share mutations with their associated melanomas varies across studies, with some reporting high concordance [8] and others suggesting that nevi and melanomas may occasionally evolve independently [12,13].

In this context, the comparative analysis of matched benign skin, nevi, and melanomas from the same patients provides a unique opportunity to dissect the genetic evolution of melanoma in vivo. Building on prior work by Dobre et al. [14] and others, who applied sensitive assays to detect low-frequency mutations in normal skin, nevi, and melanoma, we investigated 46 candidate melanoma-associated genes in a cohort of 15 patients. By integrating somatic mutation data with clinical and histopathological data, we sought to characterize the patterns of shared and private mutations across tissue types, assess the role of UV-driven mutagenesis, and identify trends in the stepwise evolution from healthy skin to nevus and melanoma. Our findings support a model of progressive mutational accumulation, highlight the heterogeneity of melanoma evolution across patients, and provide insights into the interplay between environmental exposure and genetic alterations in shaping melanoma trajectory.

## 2. Results

Fifteen patients were analyzed, with 14 contributing full matched trios (healthy skin, nevus, and melanoma) and one (patient 15) only nevus and melanoma samples. A targeted panel of 46 candidate genes was sequenced in all tissues. Given the modest cohort size, all analyses are exploratory and intended to be descriptive and hypothesis-generating.

### 2.1. Mutational Burden and Key Recurrent Genes (Cohort-Level)

Mutation burden showed a progressive increase from normal skin to nevi and then melanoma. Median mutation counts per megabase were 70.3 in benign tissue, 87.5 in nevi, and 134.4 in melanomas—representing approximately a 50% higher load in nevi relative to benign skin and an additional 47% increase from nevi to melanoma (Figure 1). Although these differences did not reach statistical significance, they describe a pattern consistent with stepwise accumulation of somatic alterations.

The most frequently mutated genes in normal skin were *FAT1*, *NOTCH2*, *NOTCH3*, *FAT2*, and *PTCH1*, mainly with largely non-canonical non-driver mutations, consistent with previous reports of UV-exposed normal epidermis [9,10]. Nevi most often harbored canonical MAPK-pathway mutations in *BRAF* and *NRAS*, as well as in *GRIN2A*, while melanomas, along with *BRAF* and *NRAS*, frequently carried additional driver alterations in *ARID1A*, *ARID2*, *CDKN2A*, and *PTEN* (Table 1, Figure 2). These distributions descriptively reflect progressive enrichment of known melanoma-associated genes.

### 2.2. Mutations in Driver Genes

Driver mutations were common across samples: 10/14 healthy skin samples, 12/15 nevi, and 14/15 melanomas harbored more than one mutated driver gene. *BRAF* was the most frequently altered driver gene, present in 7 nevi and 10 melanomas, consistently through the canonical V600E hotspot [5]. Other recurrent drivers included *FAT2*, *ARID2*, *GRIN2A*, and *FAT1* (Figure 3). These frequencies are reported descriptively and should not be interpreted as definitive estimates of prevalence.

### 2.3. Association of VAFs with Different Mutation Types

Variant allele frequencies (VAFs) differed by functional class (Table 2). Median VAFs were higher for canonical driver mutations (0.151) than for non-driver nonsynonymous (0.145) or synonymous mutations (0.112). The ratio of synonymous to nonsynonymous mutations was highest in healthy skin, lower in nevi, and lowest in melanoma, consistent with a descriptive gradient from neutral background mutagenesis to increasing representation of functional variants. In contrast, nonsynonymous variants—including non-drivers—rose progressively from normal to nevus and then to melanoma, reflecting increasing selective retention of functional alterations. This trend was strongest for canonical driver nonsynonymous mutations, which showed the largest enrichment from benign skin to melanoma. These patterns indicate greater clonal representation and positive selection of oncogenic variants during tumor evolution. Although not statistically significant in this small cohort, the VAF hierarchy supports selective expansion of driver alleles relative to synonymous/passenger mutations.

### 2.4. UV-Associated Mutations

C>T and G>A transitions—canonical UV-signature changes [15]—represented 53.7% of all substitutions. UV-type changes were more prevalent among synonymous than nonsynonymous mutations (68% vs. 50.5%) (Table 2), consistent with an origin in background UV damage that does not confer clonal advantage [9]. UV-enrichment did not reach significance in the overall Fisher’s exact test (although it was close, OR = 1.74, *p* = 0.08), likely reflecting limited statistical power due to sample size. Tissue-specific analyses showed similar trends in healthy skin and nevi (Table 2), with reduced UV enrichment in melanoma, indicating increasing contribution of non-UV mutagenic processes or selective trimming of UV-type changes during malignant progression.

### 2.5. Concordance Between Matched Nevus and Melanoma

Nine of fifteen patients shared at least one identical mutation between nevus and melanoma (Figure 2). Eight patients had shared (identical) activating canonical MAPK driver mutations (Table 3): *BRAF* V600E in six patients (P01, P02, P06, P08, P09, and P10) and *NRAS* Q61R/K in two (P03 and P15). These drivers were absent from adjacent healthy skin. VAFs were consistently higher in melanoma than in matched nevi among these eight cases, descriptively indicating clonal expansion during progression. While consistent with a nevus-to-melanoma evolutionary relationship, this observation does not exclude alternative evolutionary scenarios.

### 2.6. Tumor-Private (“Progression”) Events

Several melanoma samples harbored mutations not present in the matched nevus. These melanoma-private mutations occurred in genes linked to chromatin regulation (*ARID1A* and *ARID2*), cell growth (e.g., *PIK3CA*), cell adhesion (e.g., *EPHA2*), splicing (*SF3B1*), angiogenesis (*PTPRB*), and classic tumor suppressors (*NOTCH3*, *CDKN2A*, and *PTEN*). These private events—particularly in *PIK3CA*, *EPHA2*, *TP53*, *CDKN2A*, *PTEN*, and *ARID1A*—may support acquisition of additional hits beyond early MAPK activation. These melanoma-private variants may represent later events, although their absence in nevi may also reflect limited sensitivity to low-VAF subclonal mutations or sampling constraints. However, to our knowledge, mutations in these genes have not been reported in nevi. Overall, these patterns reflect late progression events and align with the multistep model of melanomagenesis.

## 3. Discussion

The progressive increase in mutation burden from benign skin to nevi and melanoma, although not statistically significant here, aligns with established models of melanoma evolution [8], though given the cohort size, these findings should be interpreted as descriptive rather than conclusive. Our observed ~50% increase in nevi relative to benign skin and ~47% increase from nevi to melanoma replicate earlier large-scale genomic studies showing sequential accumulation of somatic alterations from precursor to malignant states [8,16]. This trend stresses the contribution of mutation accumulation to melanocytic transformation. Importantly, because several observed trends did not reach statistical significance, the present findings should be interpreted cautiously and viewed primarily as exploratory signals that warrant validation in larger cohorts rather than as definitive evidence of stepwise mutational progression.

A subset of patients deviated from this pattern, exhibiting similar or even lower mutation counts in melanoma relative to their nevi. Such cases may reflect melanomas arising through less mutation-dependent mechanisms, or through copy-number changes, structural variation, epigenetic alteration, *TERT* promoter mutations, or mutations outside the panel’s coverage. These exceptions highlight evolutionary heterogeneity in melanocytic neoplasia and indicate that multiple routes to malignancy coexist.

The comparison of synonymous and nonsynonymous variants across tissues illustrates the balance between neutral mutagenesis and selective expansion. Synonymous mutations displayed lower and more variable VAFs, consistent with their role as largely unselected passengers, particularly those induced by UV exposure. This pattern matches previous findings in normal skin and nevi, where widespread UV-type mutations accumulate at low VAFs without phenotypic impact [9,10]. In contrast, nonsynonymous variants demonstrated higher VAFs, suggesting selective preservation or expansion during nevus and melanoma progression. Importantly, canonical driver mutations showed the highest VAFs—exceeding both synonymous and non-driver nonsynonymous variants in nevi and melanoma—supporting strong positive selection for MAPK-pathway activation and other functional alterations during malignant transformation. This pattern is consistent with, although does not prove, positive selection of functional mutations.

The decreasing relative contribution of UV-type substitutions in melanoma further supports the notion that, while UV exposure seeds the mutational landscape, later evolutionary stages are shaped by additional mutational processes and selective pressures [3,4,5,6,8]. Our findings therefore reinforce the model in which early UV-induced diversity provides the substrate for clonal selection of driver alleles such as *BRAF*, *NRAS*, *ARID1A*, and *ARID2*. However, these interpretations are constrained by the absence of copy-number, structural, and promoter-level analyses.

More than half (53.3%) of nevus–melanoma pairs shared identical MAPK driver mutations, consistent with prior sequencing studies showing that benign nevi frequently harbor *BRAF* or *NRAS* mutations and that a substantial fraction of melanomas retain the same driver mutation as their precursor [8,14,17]. Higher VAFs in melanomas relative to matched nevi highlight clonal expansion of MAPK-mutant populations and support their role as early initiating events, with later pathogenic alterations in genes such as *TP53*, *CDKN2A*, *PTEN*, *ARID1A*, and *ARID2* contributing to full malignant transformation. These data strengthen the shared-lineage model of melanomagenesis and demonstrate how VAF-based quantification reveals clonal dynamics.

Detection of somatic mutations—including canonical drivers—in healthy skin, aligns with recent evidence that sun-exposed epidermis contains clones harboring cancer-associated mutations, with clone size influenced by age and UV exposure [9,10]. The low VAFs of these mutations in our samples suggest limited clonal expansion, consistent with their early, largely neutral behavior. A pilot ddPCR study [14] similarly found low-level *RAS/BRAF* variants detectable in normal skin of patients in whom nevi and melanomas shared the same MAPK driver.

This study has several limitations inherent to the use of a focused targeted sequencing panel and single-region sampling of paraffin-embedded (FFPE) tissues. The restricted panel breadth prevents detection of broader genomic alterations, including copy-number changes, structural variants, *TERT* promoter mutations, and mutations in genes outside the targeted set, which may contribute to melanoma progression. Similarly, single-region microdissection may not fully capture intra-lesional heterogeneity, particularly in nevi where subclonal driver or cooperating mutations can be spatially restricted. These constraints could lead to underestimation of true mutational diversity and may partially explain cases in which nevus and melanoma samples appeared discordant—defined here as the absence of shared somatic variants between paired nevus and melanoma samples. Despite these limitations, the study demonstrates that key evolutionary features can still be detected using clinically routine materials.

Most studies defining nevus–melanoma evolutionary relationships employed whole-genome or whole-exome sequencing on fresh specimens or extensive multiregional microdissection [7,8,17,18]. The present study demonstrates that comparable evolutionary signals—stepwise increases in mutational burden and clonal expansion of MAPK drivers—can be detected using routine FFPE archival tissue and a focused targeted panel. Because FFPE archives and targeted NGS panels are widely available in clinical settings, this approach is scalable and cost-effective, enabling retrospective analyses across large pathology collections and facilitating translational studies linking molecular evolution to clinical outcomes. Thus, our findings extend the translational applicability of WGS/WES studies by showing that clinically feasible workflows can capture key evolutionary features of melanocytic progression.

A second important observation is the presence of discordant mutations between nevus and melanoma in a subset of patients, showing that not all melanomas arise from the immediately adjacent nevus. This heterogeneity is supported by the previous literature proposing multiple evolutionary trajectories, including *de novo* melanoma development [19,20]. Discordance may also result from mechanisms not captured by targeted panels, including copy-number alterations and structural variants, which often emerge later in melanoma development. Additionally, these and similar studies [19,21] show intra-nevus subclonal heterogeneity—including spatially restricted low-VAF or TERT-promoter mutations—implying that a melanoma may arise from a minor nevus subclone not sampled in routine microdissection [7,8,17]. Technical factors (e.g., single-region sampling, and panel breadth) further contribute to discordance. Thus, discordant mutation profiles do not prevent a nevus origin but rather highlight the existence of multiple evolutionary pathways—including nevus-first, de novo, and heterogeneous intra-nevus progression. Although our data supports this diversity descriptively, due to small sampling size, it does not allow quantification of relative frequencies.

## 4. Materials and Methods

Fifteen melanoma patients were included in the study, of whom fourteen provided complete matched tissue trios consisting of healthy skin, nevus, and melanoma, while one patient (patient 15) contributed only nevus and melanoma samples. All fifteen patients were recruited in the Department of Dermatology of the Valencia General University Hospital, from a subset of patients that had developed a melanoma adjacent to a nevus.

In this study, ‘healthy normal skin’ is used to denote histologically normal, sun-exposed skin adjacent to the nevus or melanoma. While clinically unremarkable, these samples are not equivalent to mutation-free normal controls and may contain low-level UV-associated somatic alterations, as previously described in sun-exposed epidermis [9,10]. Moreover, these samples contain a mixture of epidermal cell types, with melanocytes representing a minor cellular fraction. As a result, detected variants likely reflect mutations present in a broader epidermal field rather than exclusively melanocyte-specific alterations. This context is particularly relevant when interpreting shared or unique variants across tissues, as field cancerization in chronically sun-exposed skin may lead to low-VAF mutations detectable in adjacent epidermis without implying direct clonal lineage between normal skin, nevi, and melanoma [9].

DNA was extracted from formalin-fixed, paraffin-embedded (FFPE) tissue sections obtained from dermatopathological archives. For each patient, three distinct regions were carefully dissected from hematoxylin–eosin-stained slides under dermatopathologist supervision, corresponding to normal (healthy) epidermis, nevus tissue, and melanoma tissue. Genomic DNA was isolated using standard deparaffinization with xylene and ethanol washes, followed by proteinase K digestion and purification with a silica column-based extraction kit (e.g., QIAamp DNA FFPE Tissue Kit, Qiagen, Venlo, The Netherlands), according to the manufacturer’s instructions. DNA quality and concentration were assessed by spectrophotometry and fluorometric quantification prior to library preparation.

All available samples were subjected to targeted deep sequencing of a 46-gene candidate panel encompassing key genes implicated in melanocytic tumorigenesis. A custom bait capture was designed using NimbleGen SeqCap EZ (Roche, Basel, Switzerland) in order to target the exonic regions of the selected genes. The total size of the targeted regions was 0.32 Mb.

The 46 genes selected for deep sequencing are *ADAM29*, *ADAMTS18*, *ARID1A*, *AR-ID2*, *BAI3*, *BRAF*, *CDKN2A*, *CRNKL1*, *EPHA2*, *EZH2*, *FAT1*, *FAT2*, *FGFR3*, *GRIN2A*, *GRM3*, *HRAS*, *IL7R*, *KMT2B*, *KRAS*, *MECOM*, *NF1*, *NOTCH1*, *NOTCH2*, *NOTCH3*, *NRAS*, *PIK3CA*, *PLCB1*, *PPP1R3A*, *PPP6C*, *PREX2*, *PTCH1*, *PTEN*, *PTPRB*, *PTPRK*, *RAC1*, *RB1*, *RBM10*, *SALL1*, *SCN1A*, *SF3B1*, *SPHKAP*, *STAT5B*, *TERT*, *TP53*, and *ZNF750*.

Sequencing of paired-end 100 bp reads was performed on an Illumina HiSeq 2000 machine (Illumina, Inc., San Diego, CA, USA). Paired-end reads were aligned to the reference human genome (GRCh38) using the BWA-MEM algorithm with default parameters. Alignment files (BAM format) containing only properly paired, uniquely mapping reads were processed using Picard tools version 1.110 to add read groups and remove PCR duplicates. Local realignments and base-quality recalibrations were conducted using GATK (v.3.2.2).

The sequencing was carried out on each tissue sample of all 15 patients, obtaining average on-target coverage across samples of 452.52x (range 133.51–1068.41x). The DNA extraction of normal skin from patient number 15 did not yield DNA of enough quality, so this sample was finally discarded from all analyses.

To identify high-confidence somatic mutations from FFPE skin biopsies, we designed a custom filtering pipeline focused on specificity. After variant calling (applying Mutect2 in tumor-only mode), we annotated the trinucleotide context of each SNV. We removed likely germline variants using public databases (dbSNP and ExAC). COSMIC mutations were preserved. To reduce technical artifacts, we excluded all C>T mutations at CpG sites, as these are common in FFPE samples. Although C>T transitions at CpG sites can reflect genuine age-related methylcytosine deamination, they were conservatively excluded in this FFPE-only, tumor-only sequencing context to minimize formalin-induced artifacts in the absence of matched germline controls, with the understanding that this approach prioritizes specificity over sensitivity.

We also applied several quality filters, including a minimum sequencing depth of 20 reads, exclusion of variants with allelic depth (AD_2_) ≤ 3 or <5 for indels, and removal of variants with variant allele frequency (VAF) ≤ 0.01. In addition, to further minimize misclassification of germline variants in the absence of matched normal DNA, variants with exceptionally high VAFs—defined as exceeding the sample-specific mean VAF plus two standard deviations—were excluded from downstream analyses.

Mutation burden was calculated as the number of high-confidence somatic single-nucleotide variants per sample, normalized to the effective size of the targeted panel (0.32 Mb) after all quality, artifact, and germline-filtering steps.

All statistical analyses and tables were generated using R version 4.2.2. Mutation counts, VAF distributions, and mutation categories (driver vs. non-driver, synonymous vs. nonsynonymous, and UV-type vs. non-UV-type substitutions) were compared across tissue types using non-parametric tests due to the limited cohort size and non-normal data distribution. Pairwise comparisons of continuous variables (e.g., VAFs) were assessed using the Wilcoxon rank-sum or Wilcoxon signed-rank test where appropriate. Associations between categorical variables (e.g., UV-associated vs. non–UV-associated substitutions by mutation class or tissue type) were evaluated using Fisher’s exact test. All *p*-values were two-tailed, and statistical significance was defined as *p* < 0.05. Summary statistics (medians, means, and ranges) were computed for descriptive purposes.

## Figures and Tables

**Figure 1 ijms-27-00532-f001:**
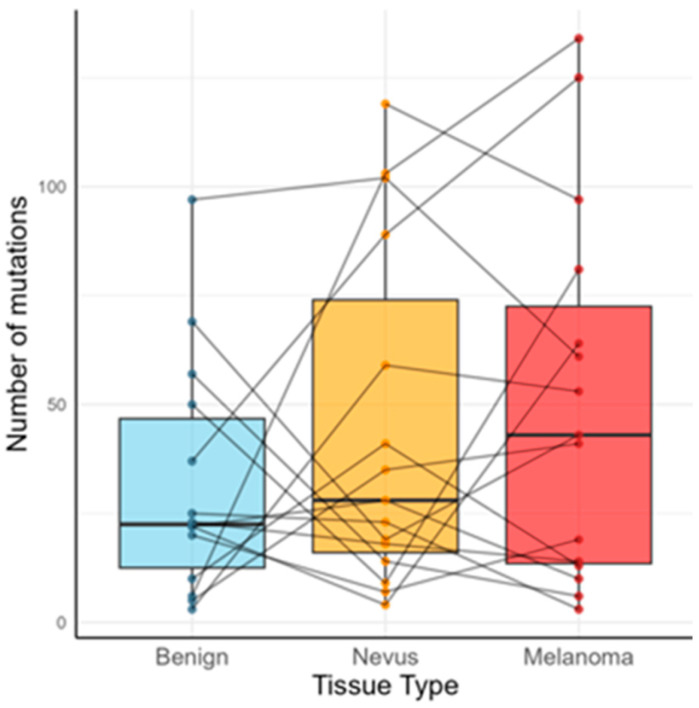
Mutational load in normal skin (benign), nevi, and melanoma: an increasing burden from healthy skin to melanoma was observed, although it did not reach statistical significance.

**Figure 2 ijms-27-00532-f002:**
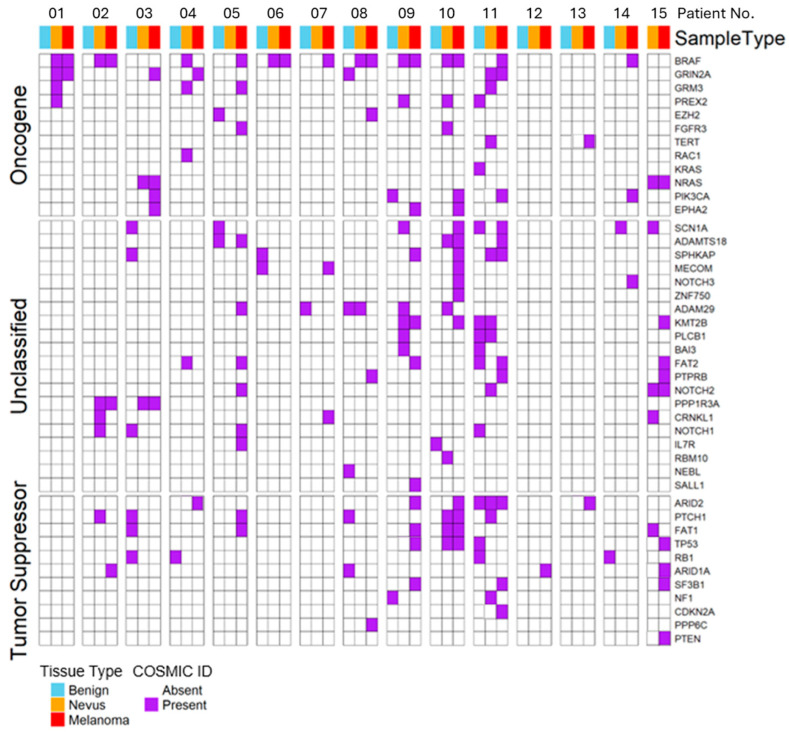
Canonical driver mutations in all 46 genes per patient and tissue type. Purple squares represent canonical driver mutations identified in the COSMIC cancer database (https://cancer.sanger.ac.uk/cosmic, accessed on 29 September 2025), displayed per patient and gene. The ‘Benign’ category refers to healthy, normal skin samples.

**Figure 3 ijms-27-00532-f003:**
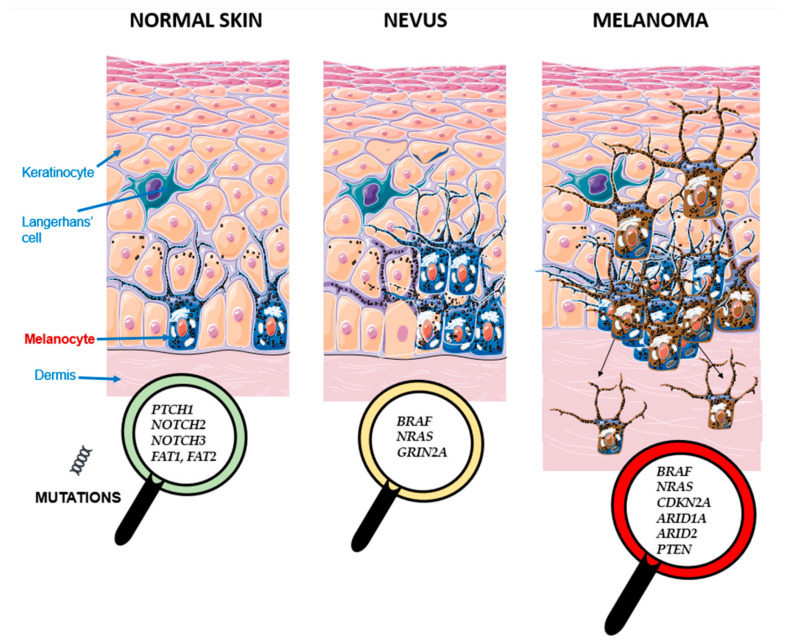
Driver genes in this study most commonly mutated during the development and progression of melanoma from healthy skin and nevi.

**Table 1 ijms-27-00532-t001:** Top 10 most frequently mutated genes (all tissues combined).

Rank	Gene	No. of Mutations	% of Total Variants	Notes
1	*BRAF*	21	8.2%	Recurrent V600E in multiple patients
2	*NRAS*	16	6.3%	Q61K/Q61R hotspot; shared nevus–melanoma lineage
3	*TP53*	14	5.5%	Found mainly in melanoma samples
4	*CDKN2A*	13	5.1%	Both p16 and p14^ARF isoforms affected
5	*ARID1A*	11	4.3%	Often truncating or frameshift mutations; in melanoma
6	*ARID2*	10	3.9%	Commonly co-mutated with *ARID1A* in melanoma
7	*PTEN*	9	3.5%	Loss-of-function variants enriched in melanoma
8	*NF1*	8	3.1%	Consistent with RAS pathway dysregulation
9	*PREX2*	7	2.7%	Known melanoma driver, higher in tumors
10	*EZH2*	6	2.3%	Missense and splicing variants in nevi and melanoma

**Table 2 ijms-27-00532-t002:** Median variant allele frequency (VAF) and UV-associated mutation proportions by tissue type and mutation class.

Tissue Type	Syn/Nonsyn Ratio	Median VAF Syn	Median VAF Nonsyn Non-Drivers	Median VAF Nonsyn Drivers	Proportion UV (Syn)	Proportion UV (Nonsyn)	*p* (UV)
Normal skin	0.31	0.095	0.128	0.133	0.71	0.53	0.16
Nevus	0.25	0.121	0.152	0.159	0.69	0.48	0.12
Melanoma	0.24	0.118	0.161	0.179	0.63	0.44	0.33
All tissues combined	0.27	0.112	0.145	0.151	0.68	0.50	0.08

**Table 3 ijms-27-00532-t003:** MAPK pathway mutations shared between nevi and melanoma in eight patients, and their corresponding VAFs.

		Activating Mutation		VAF	
Patient No.	MAPK-Pathway Mutated Gene	Nevus	Melanoma	Nevus	Melanoma
P01	BRAF	Val600Glu	Val600Glu	0.076	0.157
P02	BRAF	Val600Glu	Val600Glu	0.071	0.203
P03	NRAS	Gln61Arg	Gln61Arg	0.074	0.238
P06	BRAF	Val600Glu	Val600Glu	0.085	0.315
P08	BRAF	Val600Glu	Val600Glu	0.137	0.204
P09	BRAF	Val600Glu	Val600Glu	0.105	0.351
P10	BRAF	Val600Glu	Val600Glu	0.142	0.235
P15	NRAS	Gln61Lys	Gln61Lys	0.062	0.073

## Data Availability

The original contributions presented in this study are included in the article. Further inquiries can be directed to the corresponding authors.

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
