# Peer review of "Genetic Evolution of Melanoma: Comparative Analysis of Candidate Gene Mutations in Healthy Skin, Nevi, and Tumors from the Same Patients"

_ijms, 2026, doi:10.3390/ijms27010532_

Round 1

Reviewer 1 Report

Comments and Suggestions for Authors

This manuscript studies the mutational relationship between healthy skin, nevus, and 
melanoma using matched samples from the same patients. The study design is strong, and the 
use of paired samples is an important advantage. The sequencing depth and variant filtering are 
appropriate, and the methods are clearly described. Overall, the manuscript addresses an 
important question in melanoma biology and is suitable for publication after minor revision.
Major Comments
1. Interpretation of non-significant results
Several results are presented as trends (e.g., mutation burden, UV signature enrichment, and 
variant allele frequency differences), but these findings are not statistically significant. While 
this is acceptable for an exploratory study, some parts of the text suggest stronger conclusions 
than supported by the data.
Suggestion: Please consistently describe these findings as descriptive or hypothesis-generating 
and avoid language that implies causality or definitive conclusions.
2. Sample size limitation
The cohort size (n = 15) is reasonable for a matched analysis but limits statistical power. 
Although this limitation is mentioned, it could be emphasized more clearly.
Suggestion: Add a short statement in the Results or early Discussion noting that conclusions are 
exploratory due to the small sample size.
3. Definition of “healthy skin”
The term “healthy skin” may be confusing, as these samples represent sun-exposed skin 
adjacent to lesions rather than true normal melanocyte controls.
Suggestion: Please clarify this point explicitly in the Introduction or Methods.
4. Scope of the sequencing panel
The limitations of targeted sequencing are acknowledged, but some evolutionary 
interpretations could be more carefully framed.
Suggestion: Clearly state that copy-number alterations, TERT promoter mutations, and 
structural variants were not analyzed and may explain some discordant findings.
Minor Comments
1. Ensure consistent terminology when describing shared and private mutations across tissue 
types.
2. Minor language editing would improve clarity, especially in the Discussion.
3. Consider briefly restating the main limitations in the concluding paragraph

Author Response

Comment 1: Interpretation of non-significant results. Several results are presented as trends (e.g., mutation burden, UV signature enrichment, and variant allele frequency differences), but these findings are not statistically significant. While 
this is acceptable for an exploratory study, some parts of the text suggest stronger conclusions than supported by the data.

Response 1. We thank the reviewer for this comment. We have revised the Results and Discussion to frame these observations as descriptive and hypothesis-generating, explicitly noting the lack of statistical significance and avoiding causal or definitive language. We now emphasize that trends in mutation burden, UV-signature enrichment, and VAF differences should be interpreted cautiously in the context of an exploratory study with limited statistical power. Changes in the revised manuscript can be found in the Results section (lines 100-101; 123-124; 133-134; and 159-162), and Discussion section (lines 180-181; 205-206; and 265-267).

Comment 2. Sample size limitation. The cohort size (n = 15) is reasonable for a matched analysis but limits statistical power. 
Although this limitation is mentioned, it could be emphasized more clearly.   Response 2. We agree with the reviewer and have addressed this point in the revised manuscript. Now we emphasize more explicitly the limited cohort size and resulting constraints on statistical power in the Results and early Discussion, clarifying that the analyses are exploratory and primarily intended to generate hypotheses rather than support definitive conclusions. Changes made in the revised manuscript can be found in the Results section (lines 93-94; and 150) and Discussion section (lines 100-101).   Comment 3. Definition of “healthy skin”. The term “healthy skin” may be confusing, as these samples represent sun-exposed skin adjacent to lesions rather than true normal melanocyte controls.

Response 3. We also thank the reviewer for highlighting this point. We have clarified in the Methods and Discussion that “healthy normal skin” refers to histologically normal, sun-exposed epidermis adjacent to the nevus or melanoma, rather than distant or melanocyte-enriched control tissue. We now explicitly state that these samples contain a mixture of epidermal cell types, with melanocytes representing a minor fraction, and that detected variants likely reflect mutations within a broader sun-exposed epidermal field. Changes made in the text of the revised manuscript are the following: Materials and Methods section, lines 274-284.

Comment 4. Scope of the sequencing panel. The limitations of targeted sequencing are acknowledged, but some evolutionary interpretations could be more carefully framed.

Response 4. We also thank the reviewer for this comment. In the revised manuscript, we have more carefully framed the evolutionary interpretations to reflect the constraints of a targeted sequencing approach, explicitly noting that copy-number alterations, structural variants, TERT promoter mutations, and other genomic events outside the panel were not assessed. We also clarify that some discordant findings between tissues may reflect these alterations, or sampling limitations, rather than true absence of shared evolutionary history. Modifications in the text of the revised manuscript can be found in the Discussion section (lines 191-193; 212-213; and 233).

Minor Comments
1. Ensure consistent terminology when describing shared and private mutations across tissue 
types.

In the revised manuscript, we have standardized the terminology used to describe mutations detected across tissues, consistently using “shared” to denote identical variants present in more than one tissue type within the same patient, and “tissue-private” (or “melanoma-private”) to denote variants detected exclusively in a single tissue. This terminology is now applied uniformly throughout the Results and Discussion sections to improve clarity and avoid ambiguity. Examples of changes can be found in lines 156 and 167 of the Results section.

2. Minor language editing would improve clarity, especially in the Discussion.

Minor editing changes have been performed in the revised manuscript, including the footers of figures.

3. Consider briefly restating the main limitations in the concluding paragraph.

The main limitation has been restated in the las paragraph of the text in the revised manuscript (lines 265-267), and other sentences in the discussion section (lines 212-213; 238-239; etc.).

Reviewer 2 Report

Comments and Suggestions for Authors

The manuscript investigates melanoma evolution using targeted deep sequencing of a 46-gene panel applied to matched samples of healthy skin, nevus, and melanoma from the same individuals (15 patients). The study addresses an important biological question, and the matched-sample design represents a clear strength, as it allows intra-patient comparisons while minimizing inter-individual variability. The authors describe a progressive increase in mutational burden from benign skin to nevus and melanoma, although this trend does not reach statistical significance. They also report frequent MAPK pathway driver mutations (BRAF V600E and NRAS Q61) in nevi and melanomas, with shared driver events between paired lesions in 8 of 15 cases, supporting clonal continuity in a subset of patients. While the overall concept is compelling, there are several technical and interpretative issues that currently limit confidence in the quantitative conclusions, particularly those related to the tumor-only variant calling strategy, handling of FFPE-associated artifacts, and the definition and filtering of “mutation burden.”

  1. The pipeline uses Mutect2 in tumor-only mode and removes “likely germline” variants via public databases (dbSNP/ExAC), while preserving COSMIC. In my opinion, this approach can misclassify rare germline variants (not in dbSNP/ExAC) as somatic, inflating mutation burden and shared events across tissues. Besides, true somatic variants present in normal skin as “germline-like” depending on filtering rules and database overlap could be dismissed.
  2. Authors exclude all C>T substitutions at CpG sites as FFPE artifacts. While FFPE-related deamination is a well-recognized issue, CpG-associated C>T transitions can also represent genuine biological events, particularly those related to age-dependent methylcytosine deamination. A blanket exclusion of these variants may therefore distort mutation spectra and bias UV-signature analyses. A more robust approach would be to apply established FFPE artifact–control strategies (e.g., orientation bias filtering) and to provide sensitivity analyses showing how key results change when CpG C>T variants are retained versus excluded.
  3. The reported median mutation burdens—70.3 mutations/Mb in benign skin, 87.5 in nevi, and 134.4 in melanomas—appear unusually high for a targeted panel, even considering the depth of sequencing and the stated VAF thresholds. This raises concerns regarding variant filtering stringency and warrants clearer methodological clarification, including how mutation burden was calculated and how many variants remain after each filtering step.
  4. Although the presence of shared MAPK driver mutations provides convincing support for nevus–melanoma lineage relationships in a subset of cases, broader claims of a stepwise accumulation of mutations across disease stages are weakened by the lack of statistical significance and the relatively small cohort size. The conclusions would benefit from being more closely aligned with the statistical evidence and from avoiding over-interpretation of non-significant trends.
  5. The description of “healthy normal epidermis” requires further clarification. As melanomas and nevi originate from melanocytes, which constitute a minor fraction of epidermal cells, it is important to specify how melanocyte content was addressed in these samples. In particular, the authors should clarify whether healthy skin was obtained from adjacent sun-exposed areas or from distant sites, and discuss the potential implications of field cancerization for interpreting shared or unique variants.

Author Response

Comment 1. The pipeline uses Mutect2 in tumor-only mode and removes “likely germline” variants via public databases (dbSNP/ExAC), while preserving COSMIC. In my opinion, this approach can misclassify rare germline variants (not in dbSNP/ExAC) as somatic, inflating mutation burden and shared events across tissues. Besides, true somatic variants present in normal skin as “germline-like” depending on filtering rules and database overlap could be dismissed.

Response 1. We thank the reviewer for this comment. We acknowledge that tumor-only variant calling with database-based germline filtering is a limitation compared with matched germline sequencing and may, in principle, misclassify rare germline variants or low-level somatic variants present in normal skin. But several features of our pipeline reduce this risk. First, we applied a highly conservative filtering strategy prioritizing specificity, including strict depth and allelic depth thresholds, exclusion of known FFPE artifacts, and removal of variants present in population databases while retaining COSMIC-annotated mutations. Importantly, we also excluded variants with VAFs above a sample-specific upper threshold (mean VAF + 2 SD), which reduces the likelihood that high-VAF germline variants are retained as somatic calls. Together, these steps limit inflation of mutation burden or artificial sharing across tissues. While some rare germline variants or true low-level somatic variants in normal skin may still be misclassified or filtered out, such effects would be expected to bias results toward underestimation rather than overinterpretation. Also, rare germline variants absent from public databases would be expected to appear at high VAF across all tissues from the same patient, and that is not the case. Moreover, the central conclusions of the study rely on within-patient comparisons and on the presence of established canonical driver mutations (e.g., BRAF V600E, NRAS Q61), whose tissue-restricted occurrence and VAF patterns are consistent with somatic origin and clonal expansion. We have clarified these methodological constraints and framed all evolutionary interpretations as exploratory in the revised manuscript. Besides, we have added a new sentence to the Methods section (lines 326 to 329) in order to to further clarify this.

Comment 2. Authors exclude all C>T substitutions at CpG sites as FFPE artifacts. While FFPE-related deamination is a well-recognized issue, CpG-associated C>T transitions can also represent genuine biological events, particularly those related to age-dependent methylcytosine deamination. A blanket exclusion of these variants may therefore distort mutation spectra and bias UV-signature analyses. A more robust approach would be to apply established FFPE artifact–control strategies (e.g., orientation bias filtering) and to provide sensitivity analyses showing how key results change when CpG C>T variants are retained versus excluded.

Response 2. We thank the reviewer for raising this point regarding the exclusion of C>T substitutions at CpG sites. We fully agree that CpG-associated C>T transitions can represent genuine biological events, particularly age-related methylcytosine deamination, and that FFPE-induced deamination is not the only source of such substitutions. In the present study, however, the decision to exclude CpG C>T variants was made deliberately as a conservative strategy to minimize FFPE-related artifacts in the absence of matched germline controls. Given that our samples consisted exclusively of archival FFPE tissue and that no blood-derived normal DNA was available, we prioritized specificity over sensitivity to avoid inflation of mutation burden or distortion of clonal relationships. Several observations indicate that this stringent filtering did not compromise the biological validity of our findings. Canonical UV-associated signatures remained clearly detectable despite CpG filtering, indicating that the major UV-driven mutational signal was preserved. Also, well-established melanoma driver mutations (e.g., BRAF V600E, NRAS Q61) were robustly identified and displayed biologically coherent VAF patterns across tissues, supporting the reliability of the remaining mutation calls. Besides, the key conclusions of the study rely on relative comparisons across matched tissues within patients, rather than on absolute mutation counts or fine-grained mutational signature analyses, making them less sensitive to the exclusion of a specific substitution subclass. We acknowledge that more distinctive FFPE artifact–control approaches, such as orientation bias filtering or sensitivity analyses retaining CpG C>T variants, can be informative. However, implementing such strategies would require matched normal samples, which were not available for this cohort. Importantly, any residual bias introduced by CpG exclusion would be expected to uniformly affect all tissue types and therefore would not invalidate the comparative, hypothesis-generating nature of our analyses. We have clarified this rationale and its implications by adding a sentence in the revised Methods section (lines 319-323).

Comment 3. The reported median mutation burdens—70.3 mutations/Mb in benign skin, 87.5 in nevi, and 134.4 in melanomas—appear unusually high for a targeted panel, even considering the depth of sequencing and the stated VAF thresholds. This raises concerns regarding variant filtering stringency and warrants clearer methodological clarification, including how mutation burden was calculated and how many variants remain after each filtering step.

Response 3. The mutation burdens reported here reflect the well-established very high mutational load of sun-exposed skin and melanoma. Large genomic studies have shown that healthy normal epidermis accumulates tens to hundreds of mutations per megabase due to chronic UV exposure, often at levels comparable to cancer (e.g., Martincorena et al., Science 2015; Hernando et al., Ann Oncol 2021). Melanoma is among the most highly mutated human malignancies. In our study, mutation burden was calculated as the number of high-confidence somatic variants divided by the effective size of the targeted panel (0.32 Mb), after strict filtering to remove likely germline variants, FFPE artifacts, low-depth calls, and variants with abnormally high VAFs. Deep targeted sequencing increases sensitivity for low-VAF subclonal variants, particularly in skin, but our conclusions rely on relative, within-patient comparisons across matched tissues rather than absolute burden estimates. We have included a sentence in the Methods section (lines 330-332) in order to clarify this issue.

Martincorena, I.; Roshan, A.; Gerstung, M.; Ellis, P.; Van Loo, P.; McLaren, S.; Wedge, D.C.; Fullam, A.;, Alexandrov, L.B.; Tubio, J.M.; et al. Tumor evolution. High burden and pervasive positive selection of somatic mutations in normal human skin. Science 2015, 348, 880-6.

Hernando, B.; Dietzen, M.; Parra, G.; Gil-Barrachina, M.; Pitarch, G.; Mahiques, L.; Valcuende-Cavero, F.; McGranahan, N.; Martinez-Cadenas, C. The effect of age on the acquisition and selection of cancer driver mutations in sun-exposed normal skin. Ann Oncol 2021, 32, 412-21.

Comment 4. Although the presence of shared MAPK driver mutations provides convincing support for nevus–melanoma lineage relationships in a subset of cases, broader claims of a stepwise accumulation of mutations across disease stages are weakened by the lack of statistical significance and the relatively small cohort size. The conclusions would benefit from being more closely aligned with the statistical evidence and from avoiding over-interpretation of non-significant trends.

Response 4. We thank the reviewer for this point. We agree that, given the limited cohort size and the lack of statistical significance for several comparisons, conclusions regarding stepwise mutation accumulation should be interpreted cautiously. Accordingly, we have revised the Results and Discussion to explicitly frame these findings as descriptive and hypothesis-generating, to avoid causal language, and to emphasize that statistically supported conclusions are limited to the presence of shared MAPK driver mutations in a subset of cases. We believe these revisions align the narrative more closely with the statistical evidence and the exploratory nature of the study. The highlighted changes in the text of the revised manuscript are the following: Results section, lines 93-94; 100-101; 133-134; and 150. Discussion section, lines 180-181; and 185-188.

Comment 5. The description of “healthy normal epidermis” requires further clarification. As melanomas and nevi originate from melanocytes, which constitute a minor fraction of epidermal cells, it is important to specify how melanocyte content was addressed in these samples. In particular, the authors should clarify whether healthy skin was obtained from adjacent sun-exposed areas or from distant sites, and discuss the potential implications of field cancerization for interpreting shared or unique variants.

Response 5. We also thank the reviewer for highlighting this important point. We have clarified in the Methods and Discussion that “healthy normal skin” refers to histologically normal, sun-exposed epidermis adjacent to the nevus or melanoma, rather than distant or melanocyte-enriched control tissue. We now explicitly state that these samples contain a mixture of epidermal cell types, with melanocytes representing a minor fraction, and that detected variants likely reflect mutations within a broader sun-exposed epidermal field. We further discuss the implications of field cancerization for interpreting shared or low-VAF variants across tissues, emphasizing that their presence does not necessarily imply direct melanocyte lineage continuity. Changes made in the text of the revised manuscript are the following: Materials and Methods section, lines 274-284.

Round 2

Reviewer 2 Report

Comments and Suggestions for Authors

After reading the authors' responses and changes made in the article, the manuscript meets the standards for publication as an exploratory, hypothesis-generating study with appropriate methodological transparency and cautious interpretation of results.